# COVID-19 Vaccination in Kidney Transplant Candidates and Recipients

**DOI:** 10.3390/vaccines10111808

**Published:** 2022-10-27

**Authors:** Claudio Ponticelli, Mariarosaria Campise

**Affiliations:** 1Independent Researcher, 20122 Milan, Italy; 2Department of Nephrology, Dialysis and Kidney Transplantation, Fondazione IRCCS Ca’ Granda Ospedale Maggiore Policlinico, 20122 Milan, Italy

**Keywords:** vaccination, kidney transplantation, COVID-19 infection

## Abstract

Kidney transplant candidates and kidney transplant recipients (KTRs) are at particular risk of severe complications of COVID-19 disease. In Western countries, mortality in affected hospitalized KTRs ranges between 19% and 50%. COVID-19 vaccination remains the most important measure to prevent the severity of infection in candidates and recipients of kidney transplant. However, the uraemic condition may affect the vaccine-induced immunity in patients with advanced chronic kidney disease (CKD) and in KTRs. Retention of uraemic toxins, dysbiosis, dysmetabolism, and dialysis can diminish the normal response to vaccination, leading to dysfunction of inflammatory and immune cells. In KTRs the efficacy of vaccines may be reduced by the immunosuppressive medications, and more than half of kidney transplant recipients are unable to build an immune response even after four administrations of anti-COVID-19 vaccines. The lack of antibody response leaves these patients at high risk for SARS-CoV-2 infection and severe COVID-19 disease. The aim of the present review is to focus on the main reasons for the impaired immunological response among candidates and kidney transplant recipients and to highlight some of the present options available to solve the problem.

## 1. Introduction

Kidney transplant recipients (KTRs) are at particular risk of severe complications of COVID-19 disease. In Western countries, mortality in affected hospitalized KTRs ranges between 19% and 50% [1,2,3,4,5]. Different measures have been adopted to prevent infections in KTRs, including personal hygiene, pretransplant screening, tapering of immunosuppression, and immunoglobulin replacement therapy in hypogammaglobulinemic patients. COVID-19 vaccination remains the most important measure to prevent the severity of infection and is highly recommended by Health Authorities in candidates and recipients of kidney transplant. However, the uraemic condition may affect the vaccine-induced immunity in patients with advanced chronic kidney disease (CKD). On the other hand, in KTRs, the efficacy of vaccines may be reduced by the immunosuppressive medications.

In this narrative review, the main issues of COVID-19 vaccination in transplant candidates and kidney transplant recipients are discussed, keeping in mind that according to the World Health Organization (WHO), after the original SARS-CoV-2 strain (WA1) five variants have been identified, Alpha (B.1.1.7), Beta (B.1.351), Gamma (P.1), Delta (B.1.617.2), and Omicron (B.1.1.529), with further mutations of the spike protein.

## 2. Methods

Open-access databases, including PubMed/Elsevier and other relevant sources, including government health organizations, were searched. Only the articles published in English up to 1 September 2022, were included. Searching terms were: “SARS-CoV-2”, “COVID-19”, “CORONAVIRUS,” “COVID VACCINES, “UREMIA”, “KIDNEY TRANPLANTATION”, “IMMUNOSUPPRESSION” or combinations.

## 3. Pretransplant Vaccination

The recently available anti-COVID-19 vaccines are highly recommended for candidates of solid organ transplantation.

The first two vaccines granted Emergency Use Authorizations (EUAs) are BNT162b2 and mRNA-1273, both mRNA vaccines.

Approaches to vaccine development have included protein subunits, nucleic acids (RNA and DNA), viral vectors (non-replicating and replicating), viruses (live attenuated and inactivated), and virus-like particles [6]. All of the SARS-CoV-2 vaccines approved by the World Health Organization (WHO) were developed based on a variety of approaches and have shown different levels of efficacy (Table 1).

Both inactivated and live attenuated vaccines can be given before transplantation. However, a major issue with pretransplant vaccination is represented by a poor response to vaccine due to the impaired immune response in patients with severe CKD. Kidney transplant candidates are usually affected by CKD stage 5, a condition that can impair their immune status. The main factors that contribute to an abnormal immune status in patients with CKD include retention of uraemic toxins, intestinal dysbiosis, dysmetabolism, and dialysis.

*Retention of uraemic toxins.* Reduced glomerular filtration rate (GFR) and tubular dysfunction lead to the accumulation of a large number of uraemic toxins. Several molecules in the middle molecular range, e.g., immunoglobulin light chains, retinol-binding protein, the neuropeptides met-enkephalin and neuropeptide Y, endothelin-1, and the adipokines leptin and resistin, can exert detrimental effects on several cells of the innate and adaptive immunity [15]. As a result of toxin accumulation, plasmacytoid dendritic cells counts are decreased in patients with CKD [16], and this reduction is proportional to the loss of GFR [17]. There is also a dendritic cell dysfunction [18]. The reasons for dendritic cell malfunction are unknown but they may be partially explained by their inhibited maturation and activation caused by inositol 3-sulfate accumulation in renal failure [19]. Polymorphonuclear cell dysfunction is frequent in CKD [20] and is proportional to the increasing severity of uraemia [21]. Glucose-modified serum proteins can lead to a spontaneous apoptosis of neutrophils [22], while an accumulation of leptin, spermidine, p-cresol, and free immunoglobulin light chains can impair neutrophil chemotaxis, contributing to the disturbed immune function [23,24,25]. Uraemic toxins can also affect the function of monocytes and macrophages. Even in the early phases of CKD, monocytes show altered adhesion molecule expression [26]. In more advanced phases the retention of sulphates may modify surface molecule expression, cytokine production, and function of monocytes [27]. In end-stage renal disease (ESRD) sulphates may induce monocyte differentiation toward pro-inflammatory, profibrotic macrophages, leading to chronic inflammation [28,29]. In patients with ESRD, mitogen stimulation demonstrates a pro-inflammatory phenotype of CD4^+^ and CD8^+^ T cells with increased interferon-γ and tumour necrosis factor-α. These changes are associated with increased frequency of exhausted CD4^+^ T cells and CD8^+^ T cells [30]. End-stage renal disease can also increase apoptosis of naïve T cells, resulting in T-cell depletion of CD4 and CD8 T cell compartments [31] and may contribute to the expansion of the pro-inflammatory phenotype of memory T cells, resulting in a reinforcement of the inflammatory state already present in ESRD patients [32]. The decreased number and function of T helper (Th) cells is associated with normal or elevated numbers of myeloid cells with production of inflammatory cytokines and reactive oxygen species (ROS), a condition similar to the immunological aging observed in late elderly that leads to a defective T-cell response [33]. (Figure 1)

*Dysbiosis.* There is a continuous interaction between the microbiota and the immune system. In patients with CKD, the microbiota is completely different from that of the healthy subjects and is associated with dysbiosis, a compositional and functional alteration in the microbiota [34]. The microbiota strongly influences the transcriptional programming of innate immune cells, and specific bacterial species can directly influence T regulator cells and the balance between Th1 and Th2 cells. In addition, the microbiome has been implicated in the regulation of inflammatory processes.

Dysbiosis may produce a large quantity of end products of the bacterial metabolism that work as uraemic toxins. Gut-derived protein-bound molecules include advanced glycation end products, hippurates, indoles, phenols, polyamines, and other toxins such as methionine-enkephalin [35]. During the course of CKD, p-cresyl sulphate, p-cresyl glucuronide, trimethylamine, indoxyl sulphate, and indole-3-acetic acid tend to accumulate and affect the intestinal barrier structure and function, while in the circulation they can stimulate cells of the immune system [36]. Toxins can interfere with different functions. As an example, p-cresyl sulphate might be associated with an immune deficiency status, while indoxyl sulphate would be a main responsible of inflammation [37]. In addition, dysbiosis favours an increased translocation of living bacteria and bacterial components. This process has the potential to activate innate immunity and systemic inflammation [38]. Thus, dysbiosis may result in dysregulation of the immune system and inflammatory response [39]. This particular condition may be involved in inefficacy of COVID-19 vaccines [40]. Dietary interventions and pharmacological strategies can improve microbiome dysbiosis and reduce the burden of uraemic toxins [41].

*Dysmetabolism*. In CKD, abnormal metabolic activities of the kidney can lead to abnormal production of renin, erythropoietin, and vitamin D. Stimulation of the renin–angiotensin system is a frequent finding in patients with CKD. Activated angiotensin1 receptors can induce a shift from Th1 to T-cell regulators, thus reducing the immune response [42]. Anaemia is a constant disorder in patients with kidney function impairment and is often associated with low iron levels. Hypoferremia can blunt the immune response, because T cells need iron to support their metabolism [43,44]. Serum levels of vitamin D are usually low in CKD. Vitamin D receptors and vitamin D metabolic enzymes are present in immune cells such as antigen-presenting-cells (APC), T cells, B cells, and monocytes [45]. This may explain why vitamin D deficiency is often associated with poor immune response and increased risk of infection [46,47]. Elevated serum levels of fibroblast growth factor 23 (FGF23) are common in CKD patients. Fibroblast growth factor 23 is a phosphaturic hormone that impairs the immune response by suppressing active vitamin D, through an inhibition of the 1alpha hydroxylase and a stimulation of the 24 hydroxylases [48]. FGF23 is elevated during inflammation and can further aggravate this condition by favouring the production of pro-inflammatory cytokines [49].

Inflammation may also dysregulate innate immunity and alter T-cell or B-cell maturation and differentiation [50]. Several factors may be responsible for an inflammatory state in CKD, including increased production of pro-inflammatory cytokines [51], oxidative stress [52], hyperuricemia [53], and adipose tissue dysfunction [54]. Furthermore, the anti-inflammatory properties of high-density lipoproteins are lost in uraemia. Rather, they may contribute to the systemic inflammation in uraemic patients by modulating polymorphonuclear functions [55] (Figure 2).

*Dialysis*. The removal of uraemic toxins and the contact of blood with semipermeable membranes differ with the modality of dialysis and may have different effects on the immune response. Conventional haemodialysis (HD) is not very effective in removing middle and large molecules. In hemofiltration, solutes are removed by convection and their movement depends on membrane characteristics. The hemodiafiltration combines diffusive and convective transport. In patients treated with regular HD, T-cell immune response is deficient and the interaction between the APCs and the T-lymphocyte is impaired [56]. Even a single HD can induce a significant decrease in CD8^+^ T cells [57]. The immunodeficiency status in HD patients can be attributed to both the accumulation of uraemic toxins and the continuous activation of mononuclear cells, eventually resulting in apoptosis and accelerated senescence of monocytes [58]. Hemofiltration can remove inflammatory cytokines and improve monocyte function, improving the uraemic immune dysfunction [59]. A prospective study in chronic dialysis patients during influenza season showed that patients treated with hemodiafiltration demonstrated sustained seroprotection for longer periods in comparison with patients on HD [60]. Some reports outlined that peritoneal dialysis (PD) may better protect from decrease inflammatory status and premature ageing of the immune system in comparison with HD [61,62,63]. However, these data cannot be applied to all transplant candidates, as the immune-nutritional status in PD patients is variable. Some patients on PD may show malnutrition, increased oxidative stress, inflammation, and disrupted immune status

*Response to anti-COVID-19 vaccine.* In uraemic patients waiting for kidney transplantation, multiple factors can lead to dysfunction of inflammatory and immune cells. These abnormalities can diminish the normal response to vaccination and may require a booster. Studies evaluating the response to SARS-CoV-2 vaccines in dialysis patients reported a diminished response in comparison to healthy individuals [64]. The type of vaccine may also influence the response. An mRNA vaccine (BNT162b2, Pfizer/BioNTech) is currently the only COVID-19 vaccine that has received full FDA approval for use in the United States. Another mRNA vaccine (mRNA-1273, Moderna TX, Inc., Houston, TX, USA) is also authorized for emergency use in the United States and by the European Medical Association (EMA). A third COVID-19 vaccine (ChAdOx1-S, Astra Zeneca) has been approved by the EMA. An attenuated vaccine (Ad26.COV2.S) is authorized for emergency use in the United States as a single shot and is under critical revision by the EMA.

A study on 2367 dialysis patients vaccinated against COVID-19 with mRNA (mRNA1273 or BNT162b2) or live attenuated virus (Ad26.COV2.S) showed that most dialysis patients who received mRNA vaccines seroconverted, while 33.3% of patients receiving the attenuated adenovirus vaccine did not seroconvert and another 36% had undetectable or diminished response even 28–60 days post vaccination [65]. On the other hand, if humoral response is poor, BNT162b2 vaccination may obtain a cellular response [66]. Another study in dialysis patients confirmed an inconsistent antibody response to Ad26.COV2.S, but no difference was detected in clinical effectiveness between BNT162b2 and Ad26.COV2.S in the first 6 months after vaccination [67]. Further studies should clarify whether a change in vaccine type may be suggested for those dialysis patients who fail to seroconvert.

The response to the anti-COVID-19 vaccine may be improved after two consecutive doses of anti-SARS-CoV-2 vaccines [68,69,70,71,72]. A meta-analysis reported that 84.3% of patients on HD and 92.4% of those on PD achieved seroconversion after two doses. However, the response was less than the general population. Compared with healthy controls, HD and PD patients were 18% and 11% less likely to develop antibodies after vaccination, respectively [73]. Thus, in the maintenance dialysis population two doses of anti-SARS-CoV-2 vaccines provide moderate protection against acquiring SARS-CoV-2 infection but are highly protective against severe outcomes. However, non-responders are at risk for severe COVID-19 [74].

Patients with low antibody levels after the second dose can obtain an increase in anti-spike antibody levels after a third dose, while non-responders to two doses became low responders after the third dose. Adverse events did not seem to be more common or severe after a third vaccine dose [75,76,77,78]. Thus, a third dose of the BNT162b2 vaccine can substantially increase antibody levels in patients receiving maintenance dialysis and appeared to be as well tolerated as a second dose. In 17 HD and 28 PD patients who received a three-dose regimen of the mRNA BNT162b2, followed by a fourth “booster” dose of the mRNA vaccine (BNT162b2, *n* = 43, or mRNA-1273, *n* = 2), after a median of 7.6 months after the third dose, a significant increase in anti-spike antibody titre was found. Dose four was well tolerated [79].

Another issue is the short duration of the anti-COVID-19 mRNA vaccine in dialysis patients. Studies reported that anti-SARS-CoV-2 antibody titres declined after 6 months [80,81,82], and a more recent report showed that vaccine-specific humoral and cellular immunity waned 4 months after two vaccine doses in dialysis patients [83]. More information is also needed to assess the durability of vaccine protection against new variants. These data support the recommendation of further “booster” doses in dialysis patients. Transplant candidates should be vaccinated at least 14 days before transplantation, as immunosuppression may reduce the response to vaccines [84].

## 4. Post-Transplant Vaccination

Until few years ago, vaccination in organ transplant recipients was underutilized [85]. Live viral and bacterial vaccines were not used for fear of systemic infection [86]; inactivated vaccines were administered at least six months after transplantation when the immune reactivity was less blunted by anti-rejection therapy [87]. Vaccines were often avoided because rare cases of de novo production of anti-HLA antibodies or rejection were reported [88,89,90,91,92].

The scenario changed after the outbreak of the COVID-19 pandemic. The high risk of severe morbidity and mortality in immunosuppressed individuals prompted to recommend vaccination to kidney transplant recipients. These patients are particularly susceptible to COVID-19 infection, not only because their immune response is blunted by anti-rejection therapies but also because the kidney has abundant angiotensin-converting enzyme 2 (ACE2). ACE2 is the main receptor for SARS-CoV-2 infection. Proteolytic cleavage of viral spike protein and ACE2 by type II transmembrane serine protease (TMPRSS2) favours the physical binding of ACE2 to S-protein and plays a critical role in spreading the infection of the virus [93,94,95] (Figure 3).

Both mRNA vaccines and live attenuated vaccines have been used in transplant recipients. Yet, resistance to accept vaccination may persist. An online survey on 473 kidney transplant recipients showed that 346 (73.1%) participants planned to receive vaccination, but 105 (22.2%) were undecided, and 22 (4.7%) refused vaccination [97]. Thus, many KTRs are not vaccinated against COVID-19 and are at increased risk for infection.

*Response to vaccines*. After kidney transplantation, many dysfunctions related to the uraemic status may reverse, but immunosuppressive therapy impairs humoral and cellular reactions. A poor antibody response to vaccination has been found in KTRs even after two doses of mRNA vaccine [98,99,100]. African Americans, individuals with advanced age, and patients who received full-dose antimetabolite drugs and/or depleting therapy in the year before vaccination were at elevated risk of developing low seroconversion rates after vaccination [101,102,103]. An even lower antibody response has been obtained with Ad26.COV2.S, tested in few transplant patients [104]. Although the response to neutralizing antibodies is correlated with viral load and better outcome in patients with severe disease, vaccines might exert some protection even in patients with low levels of neutralizing antibodies, since other immune effector mechanisms including T cells and innate immune mechanisms may exert some protection [105]. Indeed, CD4^+^/CD8^+^ T-cell immunity can be detected even in the absence of seroconversion [106]. However, in spike-specific T helper cell responses, frequencies can be significantly reduced in KTRs compared with those in controls and dialysis patients, and this is accompanied by a broad impairment in effector cytokine production [98]. To define the efficacy of COVID-19 vaccination can be difficult and depends on the established end points. The conclusions may be different according to the chosen outcome measure, i.e., absence of positivity, asymptomatic disease, need of hospitalization, and/or death [107].

*Booster vaccine doses.* With the antibody response to two doses of mRNA vaccines being low, a third administration has been recommended to organ transplant recipients. In a French study, among 73 kidney transplant patients whose serum did not neutralize SARS-CoV-2 in vitro after two doses of vaccine, 14 (19%) responded after a third dose of BNT162b2 vaccine. Short time from transplantation and high maintenance immunosuppression resulted to be detrimental factors for the response to the third dose in univariate analysis. The presence of anti-SARS-CoV-2 antibodies or cellular response after the second dose predicted a response to the third dose [108]. Further studies reported rates of seroconversion ranging between 39% and 49% in infection-naive transplant recipients. Patients with low antibody titres at baseline were more likely to respond to the third dose [109,110,111,112]. The humoral response against SARS-CoV-2 in persons with a history of COVID-19 infection was greater than the response in previously uninfected participants [113,114]. Seroconversion was often associated with significant changes in cellular immunity [115,116,117]. However, the immunity to the Omicron variant is low and less than to Delta in the general population [118], and in transplant recipients the percentage of positivity against Omicron variant was low, about 18%, even after the third dose [119]. Studies with a fourth vaccine administration in kidney transplant patients reported that in spite of increased anti-spike IgG and neutralizing capacity against some variants in a few responders, transplant patients remain at high risk for Omicron [120,121,122,123]. In summary, despite booster vaccination a substantial number of kidney transplant recipients are unable to build an immune response to a COVID-19 vaccine. Emerging variants of SARS-CoV-2 are responsible of further outbreaks of COVID-19 illness.

*Immunosuppressive drugs*. As a general rule, the stronger the immunosuppressive therapy, the lower the immune response to COVID-19 vaccines. Usually, kidney transplant recipients not only lack a humoral response to COVID-19 vaccines but may also display impairment of the cellular response to SARS-CoV-2 antigens. However, some medications are more frequently associated with poor response.

Corticosteroids (CSs) have pleiotropic effects on the immune system that are time- and dose-dependent. Prednisone doses of 20 mg/day do not interfere with the immune response to inactivated vaccines [84]. The potential effect of CSs on the immunogenicity of COVID-19 vaccines has not been thoroughly investigated. The current recommendations for COVID-19 vaccines and CS administration are mostly based on the available evidence for inactivated vaccines. Ideally, vaccination should be performed when prednisone dose is ≤10 mg/day. It is still uncertain what to do in patients treated with higher doses of prednisone. However, boosters are important to maximize protection in those cases.

Mycophenolate salts are prodrugs that release active mycophenolic acid. The main mechanism of action rests on a marked reduction in guanosine triphosphate necessary for DNA synthesis and the de novo pathway of guanosine nucleotide synthesis. T- and B-lymphocytes are more dependent on this pathway than other cell types. Some investigators reported that poor response to vaccines was more frequent in kidney transplant recipients taking mycophenolate [124,125,126]. In a study, patients on mycophenolate mofetil (2 gm daily) had significantly lower SARS-CoV-2 spike-specific IgG levels as compared to patients on no or a reduced dose of mycophenolate [127]. However, it is unclear whether this is the result of a specific effect of mycophenolate or of the reinforcement of standard immunosuppression with the addition of mycophenolate.

The pharmacological activity of azathioprine rests on the formation of the active intracellular nucleotides thioinosinic acid and 6-thioguanine. Thioinosinic acid inhibits enzymes that mediate the first step of de novo pathways of purine synthesis. The administration of azathioprine may concur with other immunosuppressive drugs to impair the antibody response to two doses of SARS-CoV-2 mRNA vaccination [128].

Calcineurin inhibitors (CNI) are the mainstay of the immunosuppressive therapy in organ transplantation. Through the inhibition of calcineurin, tacrolimus and cyclosporine inhibit the synthesis of interleukin-2 and other cytokines. In comparison with healthy controls, CNI-treated transplant patients show lower humoral response and antibody titres to mRNA-vaccines [129]. On the other hand, CNI might also inhibit SARS-CoV-2 replication, at least on experimental models. Cyclophilins are required by several viruses, including SARS-CoV-2, for their replication [130].

Different than other immunosuppressive drugs, the mTOR-inhibitors, sirolimus and everolimus, can stimulate anti-SARS-CoV-2 T-cell response. Accordingly, these drugs may exert a potential beneficial role in enhancing an immune response to COVID-19 vaccine in KTRs [131].

Rituximab and anti-CD20 antibodies have limited influence on T cells, but vaccination responses can be blunted until naive B cells repopulate [132,133]. In addition, hypogammaglobulinemia may develop after rituximab [134].

Belatacept is a fusion protein that blocks the interaction between the antigen-presenting cell and the costimulatory proteins CD80-CD86. Kidney transplant recipients under treatment with belatacept showed a weak response not only to two doses [135] but also to three doses and four doses of mRNA-vaccine [136,137].

*Duration of anti-COVID-19 antibodies.* In the general population, protection against SARS-CoV-2 infection declines over time. The antibody titre usually declines around six months after vaccination [138]. A more rapid reduction has been reported in KTRs; in a study on 14 KTRs, the titre of anti-SARS-CoV-2–receptor-binding domain (RBD) antibodies dropped to 54% at 28 days and 87% at 100 days [139].

*How to improve antibody response.* Some studies are evaluating the possibility of improving the response to COVID-19 vaccines in kidney transplant recipients. In Germany, 29 patients who could not mount an antibody response to previous vaccinations received a fourth dose of a SARS-CoV-2 vaccine. To improve their immune reactivity, mycophenolate was stopped for 5 weeks in 28 participants and azathioprine in 1 participant. Seroconversion and virus neutralizing capacity after vaccination were observed in 21 (76%) participants. All the responders and four non-responders were taking CNI. Among four patients treated with belatacept, only one responded. Together with humoral-response-specific B cells, plasmablasts significantly increased. The cellular markers of T-cell proliferation (Ki67 and programmed death cell protein1) significantly increased after booster vaccination and mycophenolate hold [117]. The National Institutes of Health (NIH) is organizing another study involving 400 adults with kidney or liver transplantation. The goal is to determine if reducing immunosuppressive therapy in the days before and after a booster dose of an mRNA COVID-19 vaccine may obtain better antibody responses to vaccination in kidney and liver transplant recipients. Participants must have no recent transplant rejection or change in immunosuppression and a negative antibody response at least 30 days after two to four doses of mRNA COVID-19. Participants will be randomly assigned to one of two groups. One group will receive an additional dose of a COVID-19 mRNA vaccine with no further intervention. The other group will take a reduced dose of their immunosuppressive therapy with tacrolimus for five days before and two weeks after receiving an additional dose of a COVID-19 mRNA vaccine. Investigators will measure the antibody response to vaccination 30 days after the additional vaccine dose. The endpoint is to determine the proportion of participants who achieve a predefined antibody response. Participants will be followed for one year after enrolment. This writer is concerned about the safety of this approach. A transient reduction in immunosuppression may be well tolerated by some transplant recipients, but others who are borderline of an operational tolerance may develop a chronic rejection in the long term.

*Side effects of vaccinations.* Reactions to COVID-19 vaccination are usually mild. Fever, headache, muscle pain, nausea, vomiting, itching, and/or joint pain are common but reversible. Anaphylactic shock rarely occurs [140]. The possibility that anti-COVID-19 vaccines may increase the risk of rejection is not substantiated by the available data [141,142]. A prospective study in 58 renal transplant recipients followed for 3 months reported that SARS-CoV-2 mRNA vaccination did not elicit a significant alloimmune response [143]. However, two cases of biopsy-proven acute rejection have been described. Rejection occurred a few days after BNT162b2, ChAdOx1, and AZD1222 vaccines. Histology showed a cell-mediated rejection in both biopsies, but in one of the two there were also deposits of C4d. Both patients responded to appropriate therapy with a partial recovery [144,145]. In the general population, several cases of interstitial nephritis or glomerulopathies have been reported after COVID-19 vaccination, including minimal change disease, focal glomerulosclerosis, IgA nephropathy, and collapsing glomerulonephritis [146,147,148,149,150,151,152]. A single case of collapsing glomerulonephritis has been seen in a kidney transplant recipient after COVID-19 vaccination [153]. In healthy volunteers, alterations in haemoglobin A1c, serum sodium and potassium levels, coagulation profiles, and renal functions have been reported after vaccination with an inactivated SARS-CoV-2 vaccine, suggesting that vaccination mimicked an infection. Relevant reduction in CD8^+^ T cells and increase in classic monocyte contents were also observed. Moreover, nuclear factor kinase B signalling and reduced type I interferon responses were documented by biological assays [154]. The low incidence of side effects after vaccination in kidney transplant recipients is strange, since these patients should be particularly susceptible to side effects in view of their compromised immune response.

## 5. Conclusions

Vaccination in kidney transplant candidates and recipients remains a major issue mainly because most patients do not respond to both vaccination and the following boosters. Almost 50% of kidney transplant recipients are unable to build an immune response, particularly if treated with antimetabolites. The lack of antibody response leaves kidney transplant recipients at high risk for SARS-CoV-2 infection and severe COVID-19 disease. Responders do not achieve effective immune protection against Omicron variants, although develop a less severe disease. Thus far, an effective measure to prevent the contact with SARS-CoV-2 remains in maintaining physical distance of at least 1 m and using a face mask and eye protection in public and health-care settings [155].

The effectiveness of injectable vaccines wanes over time, and COVID-19 variants can evade the vaccines. A means to prevent COVID-19 disease may rest on infusion of anti-SARS-CoV-2 hyperimmune globulins obtained by convalescent individuals [156].

A new option may consist in nasal vaccines. Two needle-free COVID-19 vaccines that are delivered through the nose or mouth have been recently approved for use in China and India. This type of vaccine could prevent the virus from entering the body in the first place. SARS-CoV-2 relies on its obligate receptor ACE2 for infection [157,158]. Experimental studies showed that a human recombinant soluble ACE2 variant can block early stages of SARS-CoV-2 infections in human kidney and vascular organoids [159]. Soluble ACE2 prevented internalization of the ACE2-SARS-CoV-2 complex and minimized the development of COVID-19 disease in a mouse model [160]. Furthermore, the protease TMPRSS2, which is critical for inducing the binding between ACE2 and coronavirus, represents a potential target for antiviral intervention. The entry of SARS-CoV-2 might be blocked by protease inhibitors [161]. Finally, another approach deserving mention is the administration of protective monoclonal antibodies such as Ronapreve^®^ (casirivimab and imdevimab) and Evusheld^®^ (Tixagevimab and Cilgavimab). These two drugs consist of two monoclonal antibodies that can be used [162] either for prophylaxis or treatment.

In conclusion, COVID-19 vaccinations provided certain protection among patients with impaired immunological response, such as candidates and recipients of kidney transplant, by reducing mortality and gravity of infection. Furthermore, the newly available antiviral drugs contribute to a better management of the infection itself. However, the pandemic is not over yet and all the innovative approaches that will be available are welcome for future prevention and treatment of this severe infection in frail populations.

## Figures and Tables

**Figure 1 vaccines-10-01808-f001:**
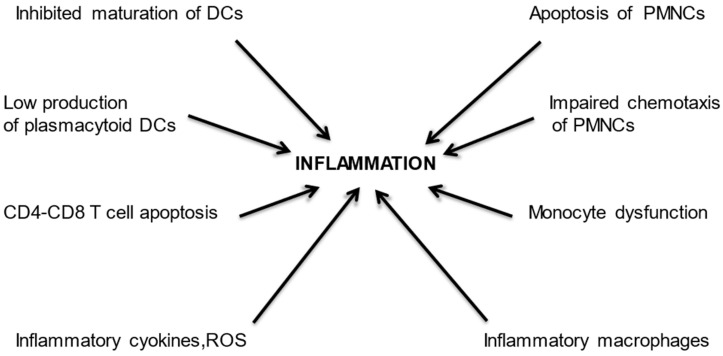
Accumulation of uraemic toxins can lead to depletion and dysfunction of dendritic cells (DCs), polymorphonuclear cells (PMNCs), and monocytes, eventually resulting in an inflammatory state. In addition, uraemic toxins can cause apoptosis of effector T cells, which stimulates the production of inflammatory cytokines and reactive oxygen species (ROS), further increasing inflammation.

**Figure 2 vaccines-10-01808-f002:**
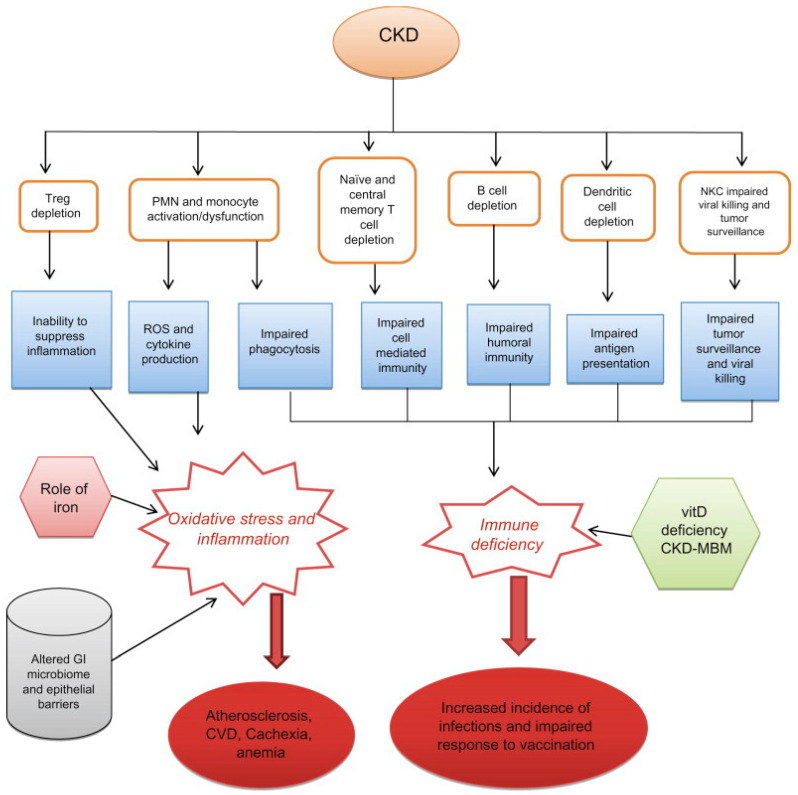
Metabolic alterations in CKD that can impair the immune response. Loss of renal function causes retention of uraemic toxins and cytokines, leading to inflammation and increased oxidative stress. The resulting pro-inflammatory uraemic milieu causes activation and decreased function of virtually all immune cells. Uraemia might also cause epigenetic changes that could result in a shift in haematopoietic stem cell populations from lymphoid to myeloid, explaining why lymphoid cell numbers are reduced. These patients also have expanded populations of circulating pro-inflammatory cells of both the lymphoid (CD4^+^CD28^−^ T cells) and myeloid cell lineages (CD14^+^CD16^++^ monocytes).

**Figure 3 vaccines-10-01808-f003:**
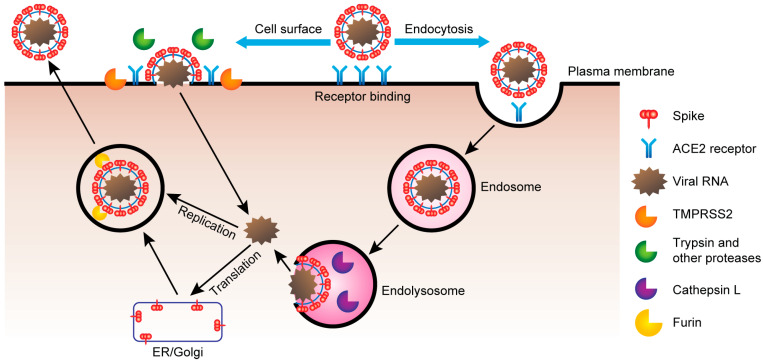
Coronavirus entry mechanisms. ACE2 is a transmembrane protein and is a main receptor for SARS-CoV-2. After binding to ACE2 the virus enters the cell where it co-opts to diverse cellular functions. Viral spike protein binds to ACE2 on responsive cells. Virus spike protein is either processed by TMPRSS2 and other serine proteases, facilitating cell surface entry, or endocytosis into endosomes, where spike is processed by cathepsin L (CTSL) in the lysosome. Viral RNA is replicated as partial and complete genome copies and translated in the ER to form new SARS-CoV-2 virions. Processing of spike protein by furin occurs prior to release of new viruses into the extracellular environment. ACE2, angiotensin converting enzyme 2; CTSL, cathepsin L; ER, endoplasmic reticulum; NRP1, Neuropilin 1; SARS-CoV-2, Severe Acute Respiratory Syndrome Coronavirus 2 [96].

**Table 1 vaccines-10-01808-t001:** WHO-approved COVID-19 vaccines.

Vaccine Name	Brand Name	Manufacturer	Approach	Dose Regimens	Vaccine Efficacy% (95% CI)
**BNT162b2**	Comirnaty	Pfizer-BioNTech	mRNA	2 doses (21 days apart)	94.6% [7]
**mRNA-1273**	Spikevax	Moderna	mRNA	2 doses (28 days apart)	94.1% [8]
**AZD1222**	Vaxzevria	AstraZeneca-Oxford	Viral vector	2 doses (28 days apart)	66.7% 55.1% (2 doses < 6 weeks apart) 81.3% (2 doses > 12 weeks apart) [9]
	Covishield	Serum Institute of India	Viral vector	2 doses (4–8 weeks apart)	~90% [10]
**Ad26.COV2.S**	Janssen COVID-19 Vaccine	Johnson & Johnson	Viral vector	1 dose	66% [11]
**BBIBP-CorV**	Covilo	Sinopharm	Inactivated virus	2 doses (21 days apart)	79% [12]
**COVID-19 Vaccine**	CoronaVac	Sinovac Biotech	Inactivated virus	2 doses (14 days apart)	50.4% (Brazil), 67% (Chile), 65% (Indonesia), 78% (Brazil), 84% (Turkey)
**BBV152**	Covaxin	Bharat Biotech	Viral vector	2 doses (28 days apart)	81% [13]
**NVX-CoV2373**	Nuvaxoid	Novavax	Protein subunit	2 doses (21 days apart)	89.7% [14]
	Covovax	Serum Institute of India	Protein subunit	2 doses (21 days apart)	90.4% (USA) 89.7% (UK and Mexico) COVOVAX

## Data Availability

Not applicable.

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
