# Peer review of "COVID-19 Vaccination in Kidney Transplant Candidates and Recipients"

_vaccines, 2022, doi:10.3390/vaccines10111808_

Round 1

Reviewer 1 Report

1. Abstract Section should be focused.

2. Motivation Section should be listed.

3. Figure 2. Metabolic alterations in CKD that can impair the immune response. Where is this Figure?

4. Nonparametric plots for the discussed data should be sketched.

5. Quotation marks should be followed.

6. More details about Figure 3 should be listed.

7. The authors sometimes wrote the abbreviation of Figure by FIG 1 or fig 1. Please use a fixed abbreviation.

8. Did the authors discussed the normality property of data?. Explain.

9. Conclusion Section should be rewritten. 

Author Response

We thank the Reviewer for his/her valuable suggestions that we tried to implement at the best of our knowledge.

Answers by point of criticism

  1. We corrected the text according to the Reviewer suggestion.
  2. We added a Method section in the text.
  3. We added the Figure 2 with more details.
  4. We did not present personal data, therefore neither the nonparametric plots nor the sketch is applicable.
  5.  
  6. We added more details to Figure 3.
  7. We corrected the abbreviation in the main text as following Figure…..
  8. Aim of this paper is not to make a systematic review of the available literature. We aimed at giving an overview of some the main reasons for the impaired immunological response among candidates and kidney transplant recipients and to highlight some of the present options available to solve the problem.
  9. We added an implemented version of the conclusion. See text.

Reviewer 2 Report

The paper “Covid-19 Vaccination in kidney transplant candidates and recipients” consists in a review of the problems accompanying and conditioning the antibody response to anti-Covid -19 vaccination in transplant patients. The review is extensive and is well structured, although the version submitted for review looks more like a draft than the final writing of the article. In particular, Figure 2 is missing and reporting the authors of the references in the text is disturbing to reading.

Some clarifications will have to be added by the authors:

1.       Vaccines approved by major regulatory bodies and used in Western countries are poorly defined in the text. Consulting the WHO page provides a better assessment of approved vaccines and approval levels. Notably in Europe, all vaccines used have been approved by emergency.

2.       The references cited by the authors are numerous: the text does not indicate the methodology used to evaluate the individual bibliographic items used.

3.       References mentioned, sometimes not matching the final numbering. (See for example 119 in the text and in the references list, and so on). To be reviewed and updated (For example, a recent review by Jiajing Li in "Transplantation" does not appear.

4.       In the findings, along with various hypotheses to prevent infection with Covid-19 and its variants, what role do the authors think specific antiviral drugs play?

Author Response

We thank the Reviewer for his/her valuable suggestions that we tried to implement at the best of our knowledge.

Answers by point of criticism

  1. We corrected according to the Review request by adding a specific section and Table 1 in the text.
  2. We added a Method section in the text.
  3. As far as the reference to the very recent paper from Jiajing li et al, we did not cite it because the paper was issued in Transplantation on Otcober 2022 and our PubMed search included papers published up to September the first.
  4. We implemented the conclusion with the answer to the Review question.